# Performance Evaluation of 5G Access Technologies and SDN Transport Network on an NS3 Simulator †

**Francesco G. Lavacca** [1,*], **Pierpaolo Salvo** [1], **Ludovico Ferranti** [1] and **Andrea Speranza** [2] and **Luca Costantini** [1]

1    Fondazione Ugo Bordoni, Viale del Policlinico 147, 00161 Roma, Italy; psalvo@fub.it (P.S.); lferranti@fub.it (L.F.); lcostantini@fub.it (L.C.)
2    Dipartimento di Ingegneria, Roma Tre University, 00146 Rome, Italy; and.speranza@stud.uniroma3.it
*    Correspondence: fglavacca@fub.it
†    This article is an extended version of paper titled *Studying and Simulation of a NS3 framework towards a 5G Complete Network Platform* and presented in the International Conference on Fixed Optics in Access Networks (FOAN) held in Sarajevo (BiH), 2–4 September 2019.

**Abstract:** In this article, we deal with the enhanced Mobile Broadband (eMBB) service class, defined within the new 5G communication paradigm, to evaluate the impact of the transition from 4G to 5G access technology on the Radio Access Network and on the Transport Network. Simulation results are obtained with ns3 and performance analyses are focused on 6 GHz radio scenarios for the Radio Access Network, where an Non-Standalone 5G configuration has been assumed, and on SDN-based scenarios for the Transport Network. Inspired by the 5G Transformer model, we describe and simulate each single element of the three main functional plains of the proposed architecture to aim a preliminary evaluation of the end-to-end system performances.

**Keywords:** 5G; SDN; LTE; vertical slice

## 1. Introduction

The progressive growth of the hyperconnected world is pushing towards the commonly named 5G (5th Generation) revolution [1], which is characterized by increasing capacity, substantial improvements of wireless systems performance, a greater capillarity and, above all, the cooperation among different systems and wireless networks. 5G networks are especially promising due to their ability to support and integrate the Internet of Things, Machine-to-Machine (M2M), Device-to-Device (D2D), Vehicular to Vehicular (V2V) communication and sophisticated logistic solutions in terms of Industry 4.0 [2–5]. Another key aspect to account for is the increase of access point density, with architectures capable of providing flexible transport solutions based on Heterogeneous Networks (HetNets) [6–8].

From an architectural standpoint, we refer to the 5G Transformer network [9] where three main planes are distinguished: Vertical Slicer (VS), Service Orchestrator (SO) and Mobile Transport and Computing Platform (MTP) (see Figure 1). VS is on the top of SO with the objective of creating customized slices and reducing the time to create services. The SO is in charge of end-to-end service orchestration and managing the transport and computing resources across one or multiple MTP. MTP represents the infrastructure, physical and virtual, over which vertical slices are created.

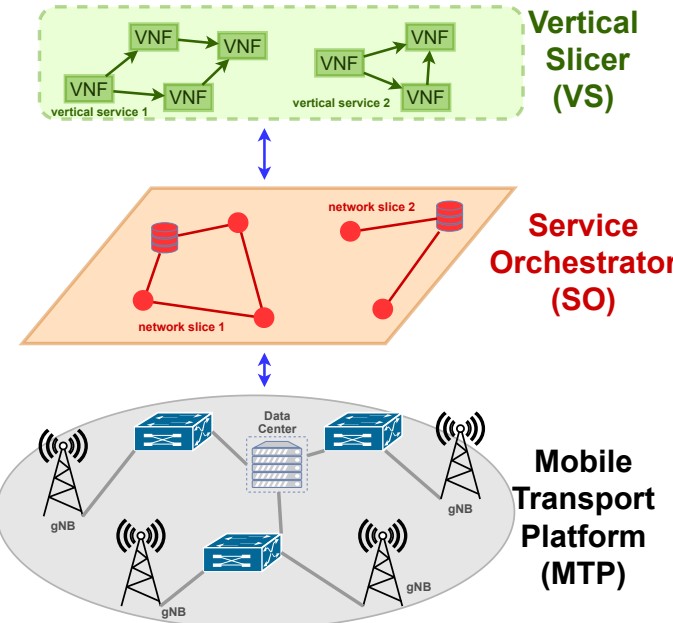

**Figure 1.** 5G transformer model.

Broadly speaking, MTP composition is two-fold: an optical architecture based on Xhaul paradigm [10] and an access network based on Long Term Evolution (LTE). In the latter, eNodeBs are connected to the Xhaul edge nodes by means of P2P optical fibers. It is important to underline that, in this preliminary work, we considered only the 5G Non-Standalone Architecture (NSA) proposed by 3GPP in Release 15, which is largely based on legacy 4G networks (LTE EUTRAN and EPC). Looking at the link capacity required from HetNets, which can range from low bit-rate (e.g., M2M and D2D applications) up to very high bit rate (i.e., mobile ultra-broadband), optical transmissions constitute a fundamental building block for wireless architectures based on the Centralized Radio Access Network (C-RAN) [11–13].

In parallel with new evolutions of network architectures, Machine Type Communication (MTC) is constantly evolving and gaining popularity day by day with a subsequent increase of demand for diverse services and applications required by vertical enterprises. As new network concepts spring, the need for networks to be more and more flexibly partitioned rises. Networks will need to satisfy specific requirements in terms of Quality of Service (QoS) bandwidth and reliability at each logical (or physical) segment. Such a flexible approach has been proposed as an innovative telecommunication strategy named Network Slicing as a Service (NSaaS) that will enable operators to offer customized end-to-end cellular networks as a service [14].

For the promised 5G revolution to happen, the whole network management will require several automatic processes to allocate a relevant amount of resources and fit user necessities while maintaining high levels of QoS, resilience and fast response times. All these requirements appear to be satisfied by architectures inspired by the well-known concepts of Software Defined Network (SDN) [15] and Network Function Virtualization (NFV) [16] technologies. Such frameworks leverage the separation of control and data planes and a central entity, which manages all network elements. This work currently features an SO based on an SDN controller managing the network elements by means of OpenFlow protocol.

Studying and profiling a complete network featuring devices belonging to each segment, from the radio access points to the core routers, could require a consistent Capital Expenditure (CAPEX) in terms of infrastructures. Furthermore, a plethora of proprietary and semi-proprietary 5G systems are still under investigation and a normalized framework is still missing. Such circumstances motivate the need for a complete simulation environment that is able to reproduce in detail all the network elements involved in the architecture and the related protocols. Both research and industry would benefit from

a common-ground simulation framework, which would help to investigate several solution aspects in terms of cost and performance. Recently, a wide interest in 5G emulation and simulation platforms based on several platforms, such as ns3 [17], GNS3 [18], NetKit [19], and Mininet [20], has spawned. Ns3 is a discrete event network simulator able to provide useful insights on the study of Internet protocols and large scale systems. Due to its close proximity to real networks' software, ns3 rises above its competitors and provides a fundamental advantage as a wide variety of real network scenarios can be simulated without the need of deploying real devices.

This article is an extended version of [21], a preliminary work on ns3 implementation of a 5G network platform. With respect to [21], we maintain the three-levels 5G Transformer model as a reference, but mainly focus on the Mobile Transport Platform with the definition of a new reference scenario. This article makes the following contributions:

- an extensive evaluation of radio access technologies in enhanced Mobile BroadBand (eMBB) scenarios;
- an in-depth study of the impact that the flexibility introduced by the SDN has in the transport segment;
- a discussion on the benefits of system integration of a network simulation able to consider complete end-to-end 5G services.

The remainder of the article is organized as follows. The reference scenario is presented in Section 2, whereas the simulated scenario is reported in Section 3. In particular, the description of the RAN is reported in Section 3.1, whilst the implemented procedures for the SDN controller are described in Section 3.2. A performance evaluation of the two network segments is carried out in Section 4. Finally, in Section 5, some conclusions and future works are drawn.

## 2. Reference Scenario

As reported before, from an architectural standpoint, we refer to a three-planes based model (5G Transformer [9]), that is roughly divided into VS, SO and MTP. Following this model, in this article, our objectives are to show the possibility of implementing a complete Mobile Transport Network in a simulation platform and to evaluate the performance of an end-to-end service at this level. As depicted in Figure 2, our reference scenario is composed of three segments: Radio Access Network, Transport Network and Core Network. In the following, a brief description of these parts is reported.

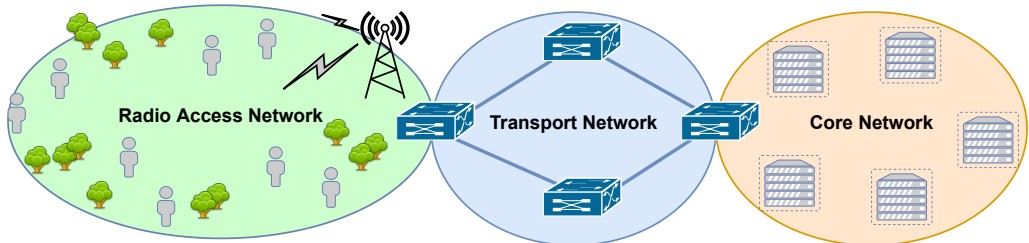

**Figure 2.** Reference scenario.

### 2.1. Radio Access Network

The green area in Figure 2 depicts the radio access segment of the considered end-to-end architecture. We assumed a "Non-Standalone 5G configuration", i.e., a configuration in which the 4G network will be the basis on which 5G technology radio access will be implemented. Details of such configuration are described in 3GPP Release 15 [22]. Simulation results are obtained through a custom version of the network simulator ns3 [17]. In particular the simulator's LTE module [23,24] implements a complete LTE cellular network architecture through two sub-models (Figure 3):

- LTE model, which includes the LTE Radio Protocol Stack (PHY, MAC, RLC, PDCP and RRC), that is installed on mobile user equipment and eNodeB and that simulates the LTE radio access;

- EPC (Evolved Packet Core) model, which includes the S1-U Protocol Stack (GTP over UDP/IP), that is installed on the network nodes of the core part of the LTE architecture (SGW, PGW and MME), and partially on the radio access nodes to the network (eNodeB); this model allows to simulate each EPC component of the cellular network and to test the IP connectivity of a mobile user connected to the Internet (for example with a simulated server connected to a network outside the LTE one).

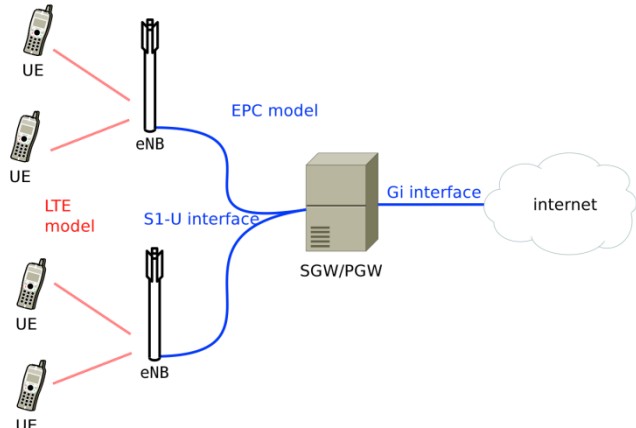

**Figure 3.** Architecture of the Long Term Evolution (LTE) system model in ns3.

To simulate the impact of peculiar 5G aspects in this network segment, we implemented in ns3 several new 5G physical layer features, inspired by the preliminary 5G New Radio indication [22]. Specifically, an implementation of eMBB services are provided and identified in the new 5G communication paradigm.

### 2.2. Transport Network

The blue area in Figure 2 represents the transport network, that interconnects the radio access nodes with the core network. An emerging network paradigm, called "Xhaul" [10], considers fronthaul and backhaul in a single segment of connectivity. Such a segment has to provide an integrated optimization opportunity, especially in sharing transport resources for different protocols and objectives. Xhaul combines and restores the traditional backhaul and fronthaul areas, by activating a flexible distribution and a reconfiguration of network elements and functions.

In this work, the transport network is based on OpenFlow capable nodes (e.g., Noviflow [25]), from now simply referred to as OF switches, and it is managed by an SDN controller. In Section 3.2, we describe in more details the procedures needed to install the flow rules on OF switches and to manage different flows.

### 2.3. Core Network

Ultimately, the area on the right side of Figure 2 depicts the core network that is represented by a number of servers in which several vertical services could be instantiated. The core network is compliant with a cloud infrastructure model that exploits the Network Function Virtualization benefits, where cloud resources hosted in different data centers could be rent [16].

In this work, the core network is out of scope. We only want to focus on radio access technologies and the closest transport network that enables 5G end-to-end services.

## 3. Simulated Scenario

In this section, we describe the implemented method and procedures in our ns3 simulator. Section 3.1 treats in detail the RAN module, whilst Section 3.2 focuses on the procedure implemented on an ns3 SDN controller that manages the transport network segment.

### 3.1. Radio Access Network

This section discusses the implementation in the ns3 simulator of the RAN supporting a particular use case that we identified within the eMBB service class, defined for the new 5G communication paradigm and implemented through vertical slicers. In this work, we select this type of new 5G service, mainly because it has a rigorous and well-defined standard. Moreover, in the near future, eMBB is expected to be the one with the most considerable impact on the transport network in terms of bandwidth.

Performance analyses are focused on below 6 GHz radio scenarios, where it has been assumed a non-standalone 5G configuration, i.e., a configuration in which the 4G network is the network architecture basis on which 5G technology radio access will be implemented [22]. Specifically, the considered RAN simulation scenario is the following: we assumed a single eNodeB located in the center of a test geographic area and a fixed number of end-users deployed in this area; each end-user needs to support high data rate services and is located in a completely random position in outdoor environments, with a propagation characteristic of Line-Of-Sight (LOS) with respect to the cellular eNodeB.

Given that the main Key Performance Indicator (KPI) for eMBB applications is the transmission band, in this work, we study the impact on bandwidth increase introduced by an eMBB scenario compared to a legacy LTE one on each single RAN access node. To this aim, we simulate different scenarios consisting of a single macrocell supporting various numbers of end-users. For each set of simulation parameters, we performed tens of simulations to account for the randomness aspects introduced by the initial deployment of end-users.

In details, the parameters used to create the RAN simulations are the following:

- a single tri-sectoral macrocell located in the center of a square simulation area is considered;
- the macrocell eNodeB radio access technology is based on a Non-Standalone 5G configuration for the eMBB scenario and on a traditional 4G network configuration for the Legacy LTE scenario;
- the implemented 5G physical layer is inspired by the preliminary 5G New Radio indication [22], i.e., the 256-QAM modulation for the DownLink and 64-QAM for the UpLink;
- simulation scenarios are composed by 1, 10 and 100 end-users randomly deployed within the simulated area;
- for each scenario, end-users start the simulation by automatically connecting to the corresponding sector of eNodeB, through idle mode cell selection procedures; during the simulation, protocols and techniques for accessing will depend on the specific radio access technology used by each considered scenario;
- end-users leverage a 15 MHz band LTE channel around the carrier frequency of 2.160 GHz in both eMBB and Legacy LTE scenarios;
- additional two elements are included in the system simulation, i.e., the SGW and PGW nodes according to the LTE network protocol, which simulates the wired elements of the cellular network architecture. These two additional network nodes are needed to measure the end-to-end performance of the communication system, in addition to the radio interface.

The simulation scenario is shown in Figure 4 where the eNodeB (red node) is connected directly to the PGW/ SGW node (green node) via a dedicated point-to-point physical connection (gray lines). To measure the network performance at the connection point between the RAN and the transport network, in this first set of simulations the PGW/SGW node has been directly connected to a second

node (yellow node), representing a network server. Finally, in order to initiate the test, an application is created on the Server node that simulates UDP traffic for each end-user (blue node) present in the eNodeB coverage area.

Finally, Figure 5 shows in detail the heat map of the radio coverage of the created RAN with 100 end-users (white spots), where it can be noticed for the 3 eNodeBs co-located in each macrocell: the propagation (in terms of SINR) of the radio signal, the orientation and the gain of the transmission/reception antennas and the relative coverage area in the simulated scenarios.

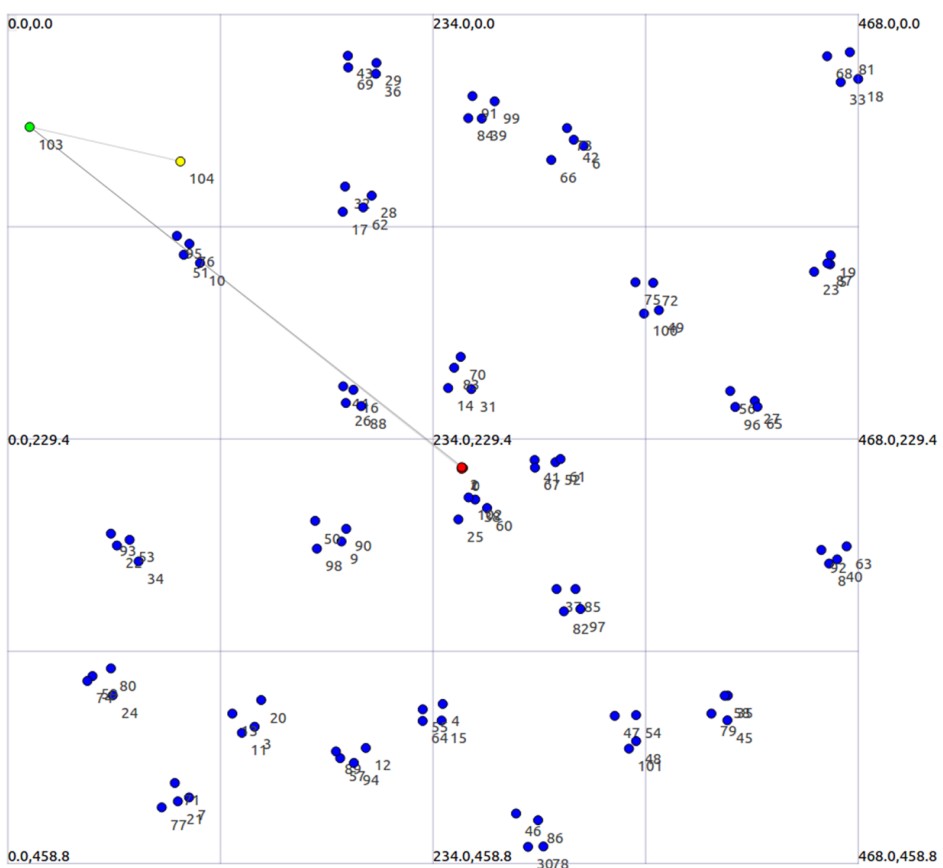

**Figure 4.** Simulated Radio Access Network (RAN) example.

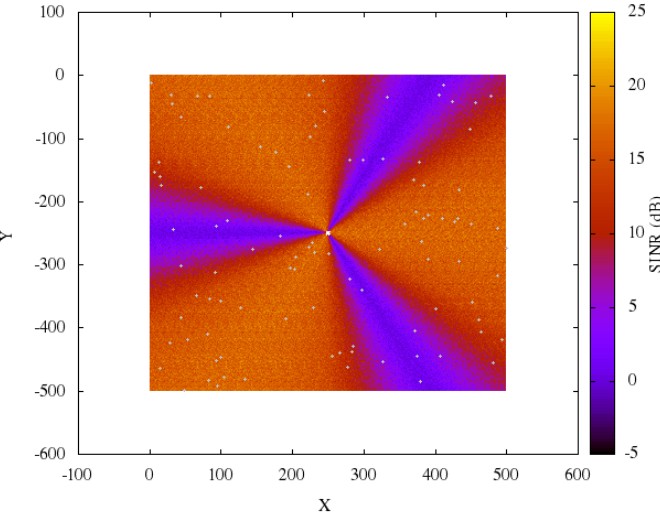

**Figure 5.** Simulated radio environment map example.

### 3.2. Transport Network

This section describes in detail how the transport network is implemented in ns3 simulator. Taking as reference Figure 2 in which is depicted our reference scenario, the transport network is based on OpenFlow capable nodes (e.g., Noviflow [25]), called OF switches, and interconnects the radio access network segment with the core network segment, that could be represented by a few number of servers corresponding to different verticals. An OpenFlow capable node is a switch device that can communicate with the controller using the OpenFlow protocol ([26]), and is able to process packets according to a generalized forwarding paradigm ([27]). The control plane of the considered network is managed by a centralized controller, in line with the Software Defined Network paradigm.

In the implemented simulator module, we consider OpenFlow 1.3 as standard for the channel communication between the SDN controller and the OF switches. In [28], authors provide a module to enhance the OpenFlow protocol version supported by ns3. Furthermore, this module provides the application interface for the switches and the controller. Whilst the OpenFlow protocol is supported and there is an application interface, ns3 does not provided a complete SDN controller. For this reason, in this work we propose and implement on ns3 SDN controller three procedures in order to route different flows following different routing policies:

- network topology discovery;
- ARP handler;
- flow routing procedure.

It is worth to noting that the implemented procedures do not represent standards, but are related to specific implementations that we done in our simulated scenario. In the following, those three procedures are described in depth.

### 3.2.1. Network Topology Discovery

In this section, we describe the procedure for the network discovery that we implemented on the SDN controller in ns3 environment.

Similar to Link Layer Discovery Protocol (LLDP), or other proprietary protocols like the Cisco Discover Protocol or Link Layer Topology Discover, the Network Discovery Protocol is part of the network management features of the SDN controller and has the objective to keep updated the topology database.

In our procedure, the topology database is implemented as a MapList, where the key is the oriented link $< tail\_swc, head\_swc >$ and the value is the output port $p$ of $tail\_swc$: thus it is possible to install a flow rule in switch A ($tail\_swc$) towards the switch B ($head\_smc$) through the output port $p$. With respect to LLDP that works at the link layer, the implemented procedure uses the transport layer to keep updated the topology database: upon a new OF switch has completed the handshake with the SDN controller, an UDP segment (EXP_msg) is generated by SDN controller with the destination port equal to 10,000 and the source port equal to $10,000 + i$, where $i$ is the ID of the OF switch. The following example describes how it is possible to obtain the topology information by means of the EXP_msg.

In Figure 6, we reported the case where switch S3 is added to the network. Upon the handshake is completed, a *EXP_msg* is generated by the SDN controller with parameters $< src\_port = 10,003$, $dst\_port = 10,000 >$ and sent to the OF switch S3 as packet-out. The switch S3 forwards this message to all output ports. When a switch receives the EXP_msg from the network, it sent it to the SDN controller as packet-in (every time an OF switch complete the handshake, the first rule installed is to forward to SDN controller the UDP messages with destination port equal to 10,000). The SDN controller leverages the features of the packet-in to add a new entry of the topology database: in particular, it learns the "tail switch" and the "port" by the packet-in header; whereas it learns the "head switch" by the *EXP_msg*. As one can notes in Figure 6a, after sending the first *EXP_msg* there are two new entries in the database.

Then the SDN controller generates two messages for the switches discovered as a tail in the previous phase with the relative source port, 10,001 and 10,002 respectively. These messages are sent

to the OF switch as packet-out and forwarded only to the port *p*. As soon as switch S3 receives these messages, they will be forwarded to the SDN controller as packet-in and other two entries will be added to the topology database. The second phase is reported in Figure 6b.

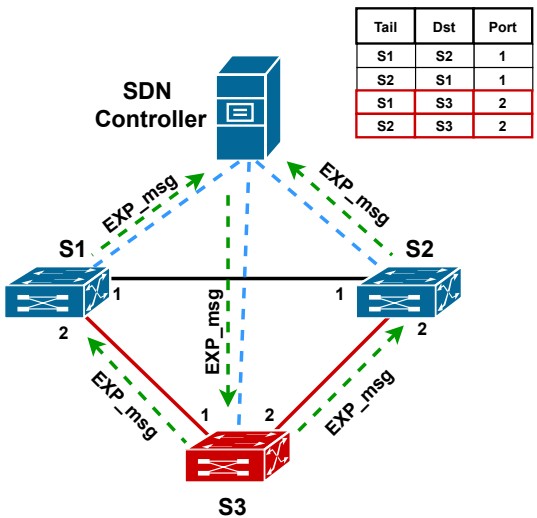

| Tail | Dst | Port |
|------|-----|------|
| S1 | S2 | 1 |
| S2 | S1 | 1 |
| S1 | S3 | 2 |
| S2 | S3 | 2 |

(**a**) Explorer message sent through the new switch.

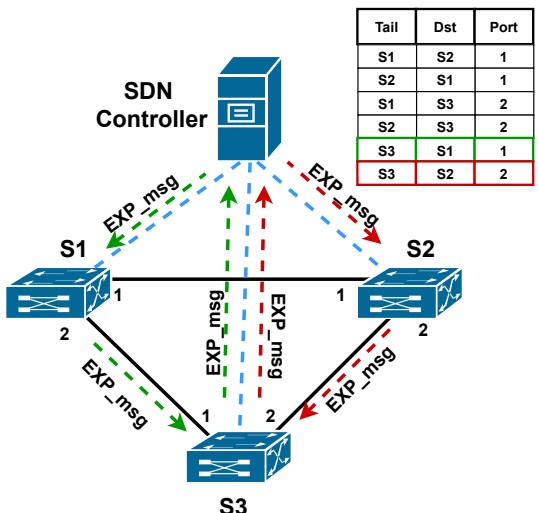

| Tail | Dst | Port |
|------|-----|------|
| S1 | S2 | 1 |
| S2 | S1 | 1 |
| S1 | S3 | 2 |
| S2 | S3 | 2 |
| S3 | S1 | 1 |
| S3 | S2 | 2 |

(**b**) Explorer messages sent through the new entries.

**Figure 6.** Example of the network discover procedure handled by SDN Controller in ns-3 environment.

### 3.2.2. ARP Handler

In this section, we describe the procedure to manage the ARP requests that we implemented on ns3 SDN controller. The aim of this procedure is to limit the number of ARP packets that are forwarded by OF switches.

The basic idea of this procedure is to allow the SDN controller to know the relationship between IP and MAC addresses of each end device. In this way, when a host wants to communicate with another device, it sends an ARP request that will be forwarded to the SDN controller by the first OF switch by means of OpenFlow packet_in messages. Here, based on the IP destination address, there are clearly two cases:

- the SDN controller knows the MAC address of the destination host;
- the SDN controller has no information about the destination host.

The former is reported in Figure 7a which represents an ARP request sent by device A to device D. If the SDN controller knows the MAC address of the destination, it directly sends the ARP reply to the source host as if it were the device D by means of OpenFlow packet_out message.

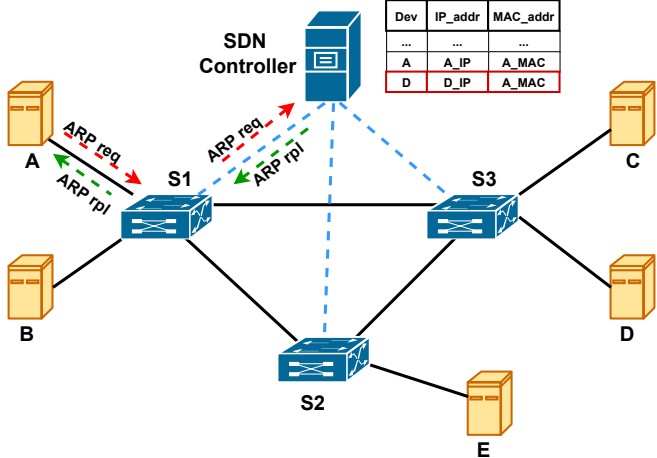

(**a**) Case in which the SDN controller already knows the receiver D.

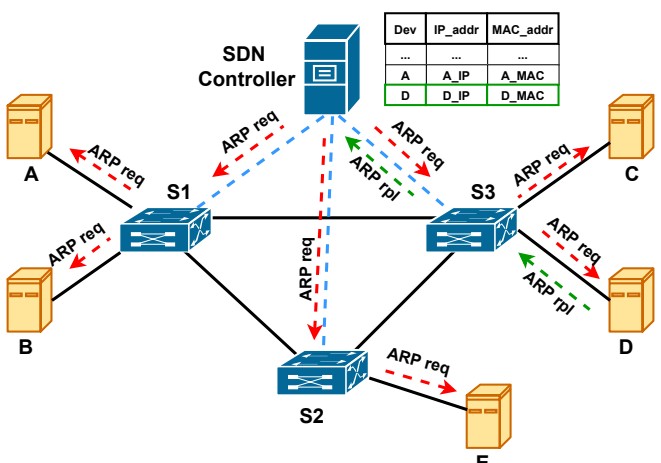

(**b**) SDN controller sends an ARP request to know the receiver D

**Figure 7.** Example of ARP request procedure handled by SDN controller in ns-3 environment.

If the SDN controller does not know the destination host, it discards the ARP request of device A and generates its own ARP request for device D. This request is sent to all switch and forwarded only to the hosts, by avoiding the OF switch ports that are configured as NO FLOOD (when the network topology is stable, the ports of an OF switch that is related to link towards other OF switches are configured as NO FLOOD. This means that broadcast messages are not forwarded on that ports). When device D receives the ARP request, it will be the only one to reply to the SDN controller. Upon received the ARP reply, the SDN controller completes its information about the device D. This case is reported in Figure 7b.

It is important to note two aspects of this procedure: firstly the SDN controller learns also about the ARP request if it has no information about the source host; secondly, by managing the ARP requests and replies, the SDN controller gets information about the closer OF switch allowing it to install the flow rules related to the hosts in the right OF switches.

Finally, it is simple to imagine that if the SDN controller is in charge to be the DHCP server, it already has information about hosts with dynamic IP configuration.

### 3.2.3. Flow Routing Procedure

In this section, we describe the flow routing procedure implemented in our simulator, which is reported in Algorithm 1. This procedure aims to obtain a capacity aggregation and thus to improve the network efficiency by considering different routing paths together.

Ip-like flow rules are installed in a pro-active manner in the flow tables of the network nodes, based on two different approaches:

- Shortest Path Routing (SPR): forwarding based on the IP destination prefix and paths computed according to a Shortest Path rule;
- Vertical-Aware Routing (VAR): forwarding based on IP source/destination address of servers, respectively for download/upload, and paths computed according to k-shortest paths and different path weights.

---

**Algorithm 1** Flow routing procedure pseudo code.

---

**Require:** A network graph: $\mathcal{G} = (\mathcal{N}^{SW}, \mathcal{L})$, current flow $fl$, table of policies $\mathcal{T}$
1: **if** $fl \in \mathcal{T}$ **then**
2:     $pl \leftarrow \mathcal{T}.weights$
3:     $n\_pl \leftarrow pl.length$
4:     $Paths \leftarrow k\_shortest\_paths(fl.src, fl.dst, n\_pl)$
5:     $Initialize(\mathcal{M}ap)$
6:     **while** $i \leq n\_pl$ **do**                    ▷ Population of $\mathcal{M}ap$
7:         **if** $pl[i]! = 0$ **then**
8:             $Curr\_Path \leftarrow Path[i]$
9:             **while** $j < Curr\_Path.length$ **do**
10:                 $rule \leftarrow\ <Port(Curr\_Path[j], Curr\_Path[j+1]), pl[i] >$
11:                 $\mathcal{M}ap.add(Curr\_Path[j], rule)$
12:                 j++;
13:             **end while**
14:         **end if**
15:         i++;
16:     **end while**
17:     **while** $y \leq \mathcal{M}ap.length$ **do**              ▷ Flow Rules Installation on OF switches
18:         $node \leftarrow \mathcal{M}ap[y].key$
19:         $rules \leftarrow \mathcal{M}ap[y].value$
20:         **while** $r \leq rules.length$ **do**
21:             $install\_flow\_rule(node, rules[r])$
22:         **end while**
23:     **end while**
24: **else**
25:     $install\_IP\text{-}dest\_rules(fl.src, fl.dst)$
26: **end if**
27: **return** $n^r, \mathcal{I}^r$

---

Three inputs are required to solve the flow routing procedure: (1) the network graph $\mathcal{G} = (\mathcal{N}^{SW}, \mathcal{L})$, composed by the set $\mathcal{N}^{SW}$ of OF switches and the set of edges $\mathcal{L}$; (2) the current flow $fl$ to be managed, that is defined by the IP source and destination address ($< fl.src, fl.dst >$); (3) the table of policies associated with the different verticals, that are defined by the tuple $< ip\_add, weights >$, where $ip\_add$ is the IP address of the vertical and *weights* is an ordered array of the weights associated to the paths. The first step is to evaluate whether the current flow $fl$ has an IP address associated with a vertical (*line 1*). In such case, the $k\_shortest\_paths$ function in line 4 evaluate $n\_pl$ different available paths for that flow. The next step is to populate $\mathcal{M}ap$ (line 6), which is a map-list where the keys are the nodes and the values are an array of rules: here *rule* is defined as a couple of output port of the node and the associated weight. Once all paths are explored and the structure $\mathcal{M}ap$ is populated,

the flow rules could be installed on the right OF switches by means of the *install_flow_rule* function. If the current flow *fl* is not checked as vertical, the *install_IP-dest_rule* function evaluates the shortest path between the source and the destination and installs forwarding rules based on destination prefix on passed through OF switches.

## 4. Simulation Results

In this section, we report the performance analysis obtained with the implemented simulator on ns3 environment. The first analysis was based on the evaluation of radio access saturation throughput by means of the new 5G technologies compared to the 4G network one. In a second step, we focused on the introduction of an SDN-based management logic in the transport segment, to evaluate different strategies apt to handle the throughput increase due to the passage from 4G to 5G. This way, a preliminary evaluation of the end-to-end system integration is inferred.

Firstly, the RAN throughput ($TH_{RAN}$) is measured at the PGW of the radio access segment (green area in Figure 2), representing the total throughput reachable by the radio access technology used in the simulation scenario.

In Section 4.1, we use $TH_{RAN}$ to evaluate the RAN data rate improvement produced by the transition from 4G to 5G.

Otherwise, we identified the Transport Network capacity ($C_{TN}$) as the maximum throughput guaranteed by the transport network segment (blue area in Figure 2) to an end-to-end service. It can be noted that this throughput is related to the capacity of the Transport Network links and, if there are different network paths, to the routing technique used to forward data flows.

It is evident that in this kind of architecture (Figure 2), the real throughput offered by the network to an end-to-end service, identified below as Service Throughput ($TH_{Service}$), is given by the following expression:

$$TH_{Service} = min(C_{TN}, TH_{RAN}).$$ (1)

Based on these considerations, we define the transport efficiency $\eta_T$ as the ratio between the Service Throughput and the RAN throughput, as follow:

$$\eta_T = \frac{TH_{Service}}{TH_{RAN}}.$$ (2)

In Section 4.2, we use this metric to evaluate the impact of the transition from 4G to 5G in the transport network segment.

### 4.1. Radio Access Technology Gain

This section provides coverage and capacity evaluations for both scenarios of eMBB and LTE. In particular, we focus our analysis on the Downlink network direction, considering a single data traffic originator, deployed outside the cellular network, and one or more destinations, located inside the cellular network domain. Service performances are evaluated in terms of the following end-users metrics:

- Throughput (TH): total received bits over to the total receiving time interval.
- Packet Delivery Ratio (PDR): ratio between the number of correctly received packets and the number of related transmitted ones.
- Delay: end-to-end delay between the transmitting and the receiving of a data packet.
- Jitter: variation of the end-to-end delay between packets belonging to the same data flow.

Additionally, network performances are evaluated in terms of the following network metrics:

- RAN Throughput ($TH_{RAN}$): Throughput measured by the eNodeB and given by the sum of the Throughput ($TH$) measured by all the end-users attached to this one.

- Radio Access Throughput Gain ($G_{RAT}$): ratio between the RAN Throughput ($TH_{RAN}$) obtained in the eMBB scenario and the RAN Throughput obtained in the legacy LTE, by simulations based on the same network parameters.

Simulations have been performed by assuming 1, 10 and 100 users randomly deployed in the area under test, for a number of independent simulation runs, and comparing the final results obtained in eMBB and legacy LTE scenarios. To estimate the effective Downlink saturation throughput reachable by the eNodeB towards one or more users, the following assumptions have been made in simulations:

- the bit-rate offered to the network layer by the application layer, is higher than the threshold of maximum traffic delivered by the network;
- ten simulation runs were carried out for each network configuration, by varying the random position of the end-users within the considered area, in order to obtain a statistic of the network performance varying the end-users positions;
- the application uses UDP as transport protocol.
- the eNodeB is able to communicate always and with all the end-users deployed in the test area (no packet loss): the simulation parameter settings, used to create the different scenarios, are reliable and adequate to avoid a non-in-coverage end-user SINR level.

Thus, a study on the RAN capacity for the two considered radio technologies was conducted.

Firstly, Figures 8 and 9 depict eMBB service evaluation results in comparison with legacy LTE for scenarios with 10 and 100 end-users. Based on 10 simulation runs varying the end-user positions, these results show mean values of throughput ($TH$), delay and jitter measured by each end-user. Moreover, the statistical validation of these results is depicted in Figure 10, showing the mean values and standard deviations averaged on all end-users in the three kind of scenario (1, 10 and 100 end-users). It is worth noting that, from each end-user's service point of view, the eMBB scenario significantly outperforms the legacy LTE in terms of every performance metrics considered, while being under the same coverage conditions.

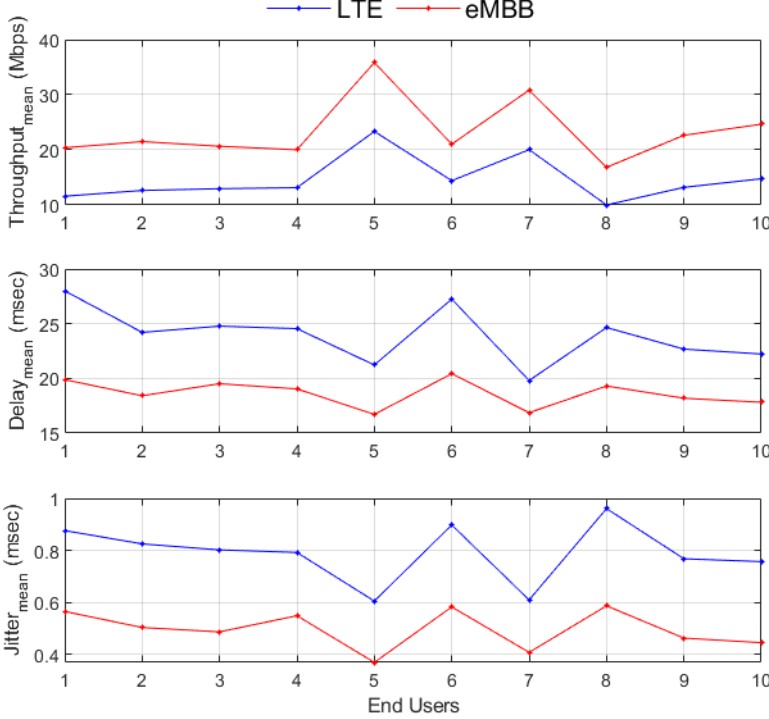

**Figure 8.** Simulation results: eMBB vs Legacy LTE services with 10 end-users.

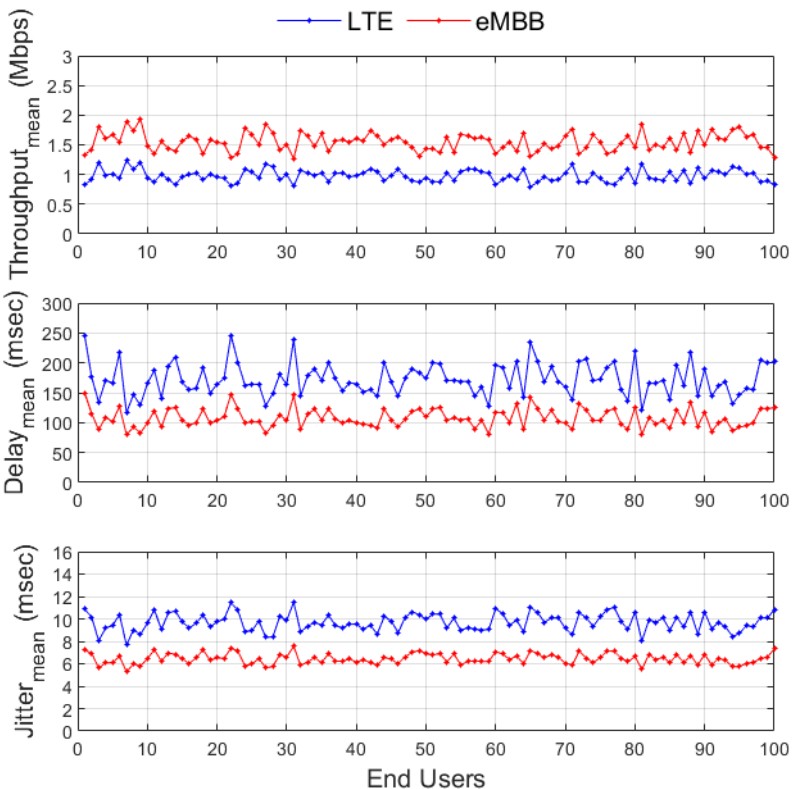

**Figure 9.** Simulation results: eMBB vs Legacy LTE services with 100 end-users.

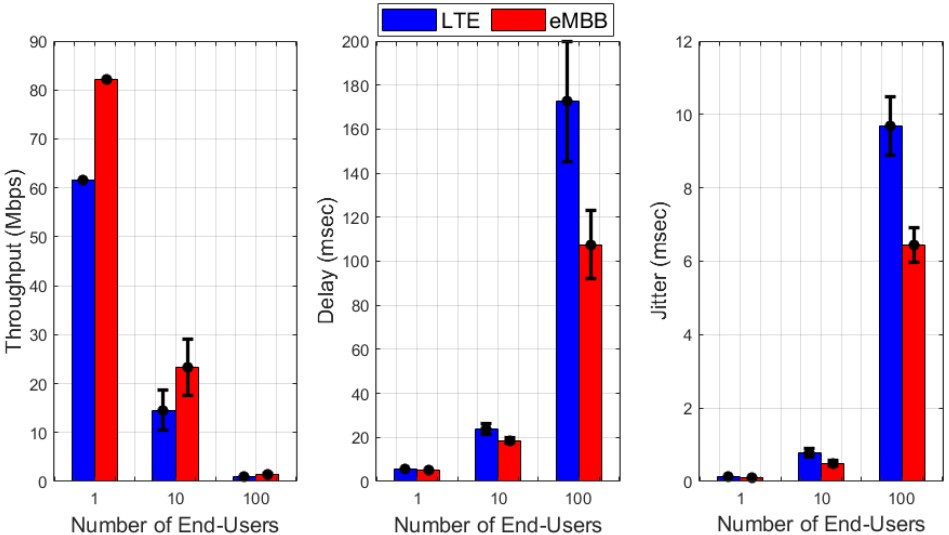

**Figure 10.** Simulation results: eMBB vs LTE services with 1, 10 and 100 end-users.

To better quantify the improvements introduced in the overall network capacity by the 256-QAM modulation in Downlink transmission, our next analysis turns to the point of view of the eNodeB, studying the performance metric called above $TH_{RAN}$.

Finally, to directly compare the considered radio access technologies, we defined the Radio Access Technology Gain ($G_{RAT}$) as the ratio between the eMBB Throughput $TH_{RAN}^{eMBB}$ and LTE $TH_{RAN}^{LTE}$, both measured by RAN simulations based on the same network parameters, as follows:

$$G_{RAT} = \frac{TH_{RAN}^{eMBB}}{TH_{RAN}^{LTE}} \tag{3}$$

Figure 11 depicts the RAN Throughput ($TH_{RAN}$) and the Radio Access Throughput Gain ($G_{RAT}$) levels obtained for different number of end-users, averaged on 10 simulation runs. In this case, we can better observe how remarkable the improvement with respect to the LTE network in scenario eMBB.

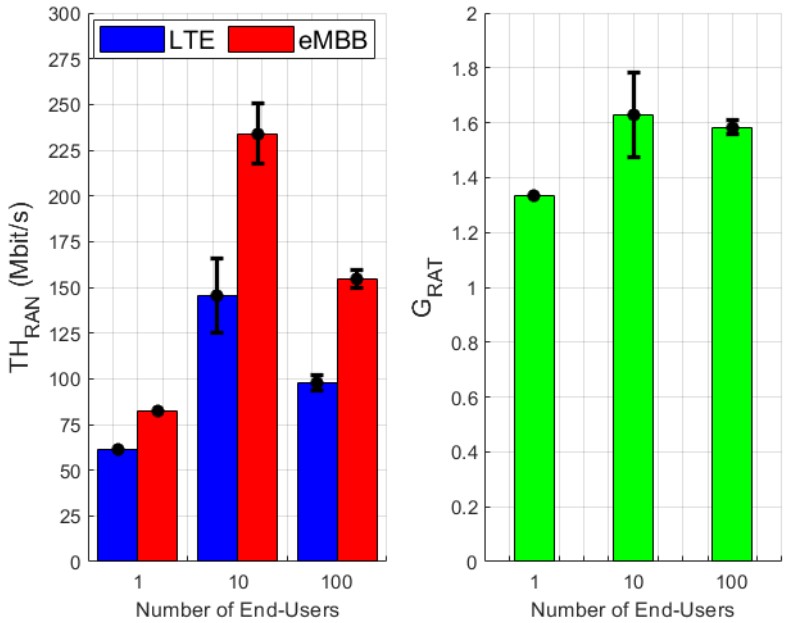

**Figure 11.** Radio Access Throughput GAIN with 1, 10 and 100 end-users.

In particular, when the RAN scenario is composed of a single end-user and the eNodeB, the technology gain from 4G Downlink to 5G is around 1.33, independently from the end-user position (standard deviation equal to zero). This number is directly related to the Downlink transmission Modulation and Coding Scheme (MCS), and so to the SINR measured between the positions of eNodeB and end-user, it can be noticed that this value exactly measures the throughput gain due to the introduction of 256-QAM modulation in downlink transmission from an end-user point of view, as in this case there is just one end-user in the scenario.

Otherwise, when the number of end-users increases (10 and 100), their downlink transmissions have to share radio resources, so, while the end-user throughput $TH$ decreases for both eMBB and LTE, as depicted going from Figures 8 and 9, the overall network throughput $TH_{RAN}$ increases respect to the scenario with 1 end-user. However, we can observe that the mean technology gain from 4G downlink to 5G is around 1.6 (right side of Figure 11) from 10 to 100 end-users scenarios, even if the corresponding measured $TH_{RAN}$ values are very different (left side of Figure 11), due to the different number of end-users sharing the common radio resources.

### 4.2. Transport Efficiency

The last analysis batch we propose is related to the evaluation of the impact of the transport network to the service throughput. The network considered is reported in Figure 12 and is composed of four OF switches and two nodes representing the service endpoints. The link capacity is equal to

150 Mbps that is the maximum RAN throughput evaluated in the previous section for the legacy LTE services. In Figure 12, three different paths are also reported. A combination of these paths is used to define three possible routing policies in compliance with the Vertical-Aware Routing (VAR) procedure:

- Alternative Paths (AP), where are considered path A and path C with equal weights;
- Load Balancing (LB), where all paths are equally considered;
- Weighted Paths (WP), where path A and path B are considered, but with double weights for B.

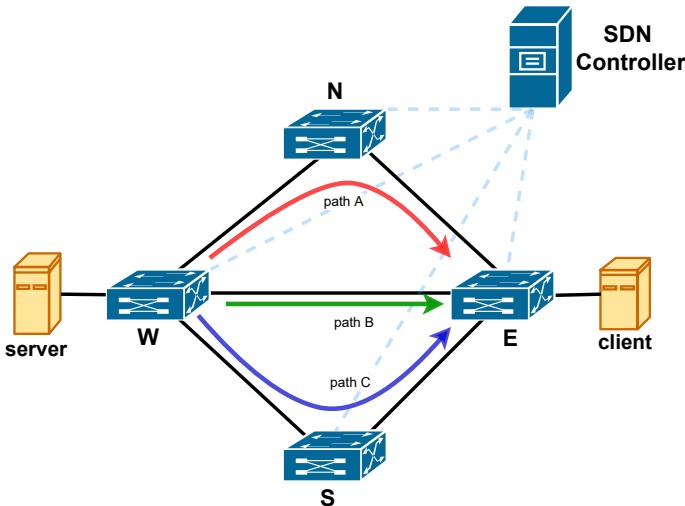

**Figure 12.** Routing paths considered in performance evaluation.

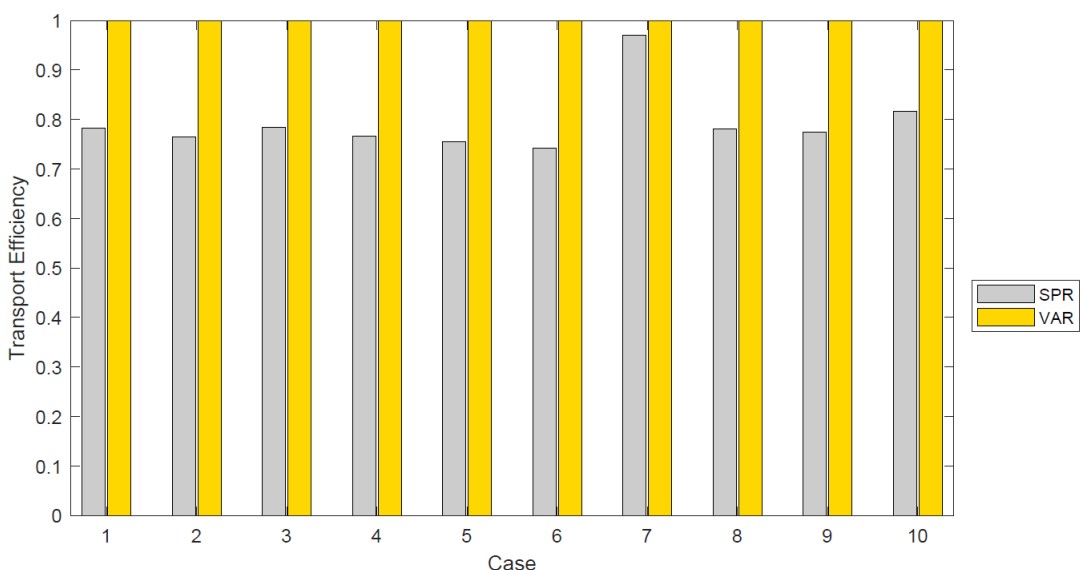

**Figure 13.** Transport efficiency.

The first evaluation of the impact of the transport network is reported in Figure 13, when two different routing strategies, shortest path routing (SPR) and VAR respectively, are considered. The transport efficiency is evaluated for the same user cases considered in the simulation carried out in Figure 8. We note that only by using the VAR strategy we could support completely the radio access throughput of the eMBB services and, obviously, obtain a transport efficiency equal to one.

Finally, the last analysis aims to evaluate the impact of three different routing strategies on the average delay and the maximum jitter for two different eMBB scenarios, respectively with 10 and 100 users. In this case, the number of users determines the RAN Throughput as reported

in Figures 8 and 9. Tables 1 and 2 show the results of this analysis. As we can notice, the AP strategy aims to perform the routing without introducing jitter. It is clear that in this case, we introduce a higher value of delay. Whereas, the LB strategy has a better delay than AP, but introducing jitter to the system. The WP is the best strategy in terms of average delay, and it has no jitter higher than the LB strategy. Moreover, tables show that the routing strategy has an impact on the average delay and the maximum jitter, thus it is important to decide for alternative paths or a load balancing strategy with respect to the vertical and the application that we have to support. For deterministic application it is more important to have a minimum jitter and constant delay; whereas delay-sensitive vertical could decide obviously for the strategy with less delay.

**Table 1.** Average delay and maximum jitter obtained for the eMBB scenario with 10 users.

| Routing Strategy | Delay ($\mu$s) | Jitter ($\mu$s) |
|---|---|---|
| Alternate Paths | 242.490 | 0.000 |
| Load Balancing | 215.155 | 54.670 |
| Weighted Paths | 189.179 | 54.879 |

**Table 2.** Average delay and maximum jitter obtained for the eMBB scenario with 100 users.

| Routing Strategy | Delay ($\mu$s) | Jitter ($\mu$s) |
|---|---|---|
| Alternate Paths | 242.490 | 0.000 |
| Load Balancing | 215.050 | 54.879 |
| Weighted Paths | 187.611 | 54.879 |

Another important thing to note is that by using different weights for the paths, we could support service without blockage and with a minimum delay, avoiding an increasing of jitter.

## 5. Conclusions and Future Works

This work deals with the enhanced Mobile Broadband service class, defined for the new 5G communication paradigm. We concentrate our study on this type of new 5G service, mainly because it has a rigorous and well-defined standard, and secondly because, in the near future, eMBB is expected to be the one with the most considerable impact on the radio access and transport networks in terms of bandwidth, given that the main KPI for eMBB applications is the transmission band.

The first proposed analysis is an evaluation of the access throughput of the new 5G technology compared to the 4G network one. Simulation results are obtained with ns3 and show the technology gain produced in the RAN, by the introducing of 256-QAM modulation in DownLink as defined in the physical layer of the 5G New Radio indication reported in [22]. In particular, we observed that the mean technology gain from 4G Downlink to 5G one is stationary to 1.33 for 1 end-user scenario, while it is around 1.6 from 10 to 100 end-users scenarios, even if the corresponding measured $TH_{RAN}$ values are very different, due to the different number of end-users sharing the common radio resources.

Then, we evaluate the impact of the SDN-managed transport network on the end-to-end service throughput. Firstly, results show that the introduction of the SDN technology can make it easier to support the increasing RAN Throughput by applying flexibly different routing strategies. These strategies exploit more network paths in spite of the link capacity is the same as the legacy LTE network. Furthermore, we demonstrate that different routing strategies have impacts of performance indexes like average delay and jitter.

Future work will concentrate on enriching the capabilities of our simulator by incorporating new 3GPP releases, with particular emphasis on millimeter-wave (mm-wave) communication.

**Author Contributions:** Conceptualization, F.G.L.; investigation, P.S.; software, A.S.; writing–original draft preparation, F.G.L. and L.F.; writing–review and editing, P.S.; supervision L.F.; project administration, L.C. All authors have read and agreed to the published version of the manuscript.

**Funding:** This research received no external funding.

**Conflicts of Interest:** The authors declare no conflict of interest.

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
