# Peer review of "Performance Evaluation of 5G Access Technologies and SDN Transport Network on an NS3 Simulatorâ€"

_computers, doi:10.3390/computers9020043_

Round 1
Reviewer 1 Report
General comments:
- I'm not a native English speaker but I think that the manuscript requires a soft proof-reading
- It is better to introduce every acronym before using it in the text
- The improvement shown in Fig. 9 and 10 should be also validated statistically
- The Conclusion section should be improved a bit. At the moment they look like an abstract.
- In the Reference section, all the entries must use the same style.
Minor comments:
- Line 26, [? ] reference is missing
- L45 well-knwn --> well-known
- L52 frameworks --> framework
- L72 as follow --> as follows
- L105 [? ] reference is missing
- L110 e.g --> e.g.
- L111 In Section 4 we describe --> In Section 4, we describe
- L131 eNodeb --> eNodeB
- L135 KPI --> Key Performance Indicator (KPI)
- L136 Lte --> LTE (check the whole manuscript)
- L138 simulation parameters we performed --> simulation parameters, we performed
- L183 ARP ?
- L269 apt ?
- Fig. 8 is not so usefull there are no differences among the columns, it can summirized in one line.
- Fig. 10 is quite hard to read
- Tab. 1 and 2 the average is computed using 10 runs, is it correct? Since it is quite weird the delay values are better for 10 users.
Author Response
We want to thank you for your usefull comments. Please see the attachment for the replies.

Reviewer 2 Report
The paper is very interesting with a lot of solid engineering and IT knowledge and quite well prepared. It may meet the audience interest. However, several parts should be developed.
The research are good example of topics evidently connected to Industry 4.0 and Logistics 4.0 subject matters - it might be add by Authors the adequate references and definitions e.g. based on: Kostrzewski et al. (2020), especially that the paper needs deep literature review.
Kostrzewski M., Varjan P., Gnap J. (2020) Solutions Dedicated to Internal Logistics 4.0. In: Grzybowska K., Awasthi A., Sawhney R. (eds) Sustainable Logistics and Production in Industry 4.0. EcoProduction (Environmental Issues in Logistics and Manufacturing). Springer, Cham, pp 243-262. DOI: https://doi.org/10.1007/978-3-030-33369-0_14
Authors are asked to give the missing reference: „MTP composition is two-fold: an optical architecture based on X-haul paradigm [? ]”, ”An emerging network paradigm, called "Xhaul" [? ]”.
Authors are asked to check the spelling, e.g. ”well-knwn”, ”Lte”, ”Ip-like”, etc.
Fig. 4. should be given in much bigger scale.
What does fig. 8 contribute to the research?
More developed conclusion and future research should be mentioned in the paper.
References are not given in a full way in the list of them, not only websites are without on-line access dates but also printed matters are lack of proper descriptions.
Author Response

(The authors gave the same response as above.)

Reviewer 3 Report
The idea of the article seems quite interesting at first, but it needs a lot of improvement. Currently, it looks just like a mix of tests and results... In particular, I have the following concerns:
- The Introduction should be rewritten. It is not ordered and it does not properly convey the idea of the authors in my opinions. References and context are ok, but they are randomly mentioned, without a proper flow. An example of this can be seen when the authors start talking about the Service Orchestrator several times (SO) and, afterwards, after mentioning several times, they suddenly explain the concept of "orchestrator" (and this should have been done way before).
The SO is currently based on a SDN controller managing the network elements by means of OpenFlow protocol
"and Software Defined Network (SDN)" <- defined afterwards!
"called Orchestrator which manages all network elements." <- where the term SO was already used several times before!
- The knowledge of the authors about SDN/NFV seems limited (and I mean "seems", but maybe it is not!). The reason for this is that they take for granted that everything is implemented in OpenFlow in these scenarios (and it is not) or, for example, they explain how SDN controllers handle ARP by stating "In this section, we describe the procedure to manage the ARP requests in a software defined network", when ARP handling might be completely different in each SDN controller! (there is not a rule or standard for this service... nor for topology discovery, etc.).
I think the authors should rather talk about the specific implementation in ns-3 (if that's the case), but not generally for SDN as they do.
A way to improve this could be merging sections 3 and 4 into a single one called "simulated scenario", because otherwise it looks like definitions or theory, but they are not.
- The presentation of the Evaluation should be improved. Authors should clearly state at the beginning of the article what they want to measure and why. Currently, they just provide a bunch of results, but this explanation is missing. I'd reduce sections 3 and 4, to provide more insights on the Evaluation.
Additionally, Figure 8 is not needed. Authors might simply explain it with one phrase and no figure.
- The writing in general is a bit poor. It looks as if the authors did not re-read the article to check errors, etc. I'm listing some examples below, though they are not the only ones:
*About acronyms: Please use the acronyms correctly. For example, if MTP was already introduced, try to avoid using "Mobile Transport Platform" without the acronym, once again. Another example is the repeated acronyms, like Vertical Slicer (VS), Service Orchestration (SO), Mobile
80 Transport Platform(MTP) or enhanced Mobile Broadband (eMBB), which are presented more than once. Finally, some terms like eNodeB, LTE and EPC (popular in the 5G field) are never introduced (though popular, they should be introduced...).
*About references: Please review the format of references (capital letters and typos like "frameworktowards").
*About grammar errors and typos:
+a SDN controller -> an SDN controller
+legacy Lte -> legacy LTE
+Ip-like -> IP-like
+X-haul paradigm [? ], "Xhaul" [? ] <- missing refs?
+titles of sections have different formats (some in capital letters, some others not...)
Author Response

(The authors gave the same response as above.)

Round 2
Reviewer 2 Report
The paper was much developed and the Authors took an effort in order to consider all the criticism and suggestions. Therefore the paper may be accepted for publication.
Author Response
Thank you again for your useful and constructive comments.
Reviewer 3 Report
The authors have addressed most of my comments (the most technical ones) and the new text looks more coherent. However, there are still some parts to check:
- I still believe the writing in general is a bit poor. The authors should consider reviewing it carefully or sending it to an English polishing service.
- An example of this are the still remaining grammar errors/typos, which I already mentioned before and were not corrected:
+a SDN controller -> an SDN controller
+X-haul, "Xhaul" -> Please, choose just one format for the same concept
+review acronyms, Radio Access Network is written many times without the acronym... and RAN as well. I think the acronym should be introduced just once and then RAN should be used the whole time.
But there are more errors, like: "in out simulation", which clearly indicates that the authors did not re-read the text to check it.
- In general, the text could be improved in organization and readability. Apart from that, technically it looks good to me.
Finally, have the authors considered publishing their implementation? In the end, as they claim, ns-3 does not have these implementations in their source code, so publishing it in GitHub (for example) could be very valuable for other researchers.
Author Response
We want to thank you again the reviewer for its useful and constructive comments that improved the technical quality of our work. Unfortunately we did the same text mistakes, but we hope to cover this weak point in this new version.
Regarding acronyms, now the expression "Radio Access Network" appears only two times in the text, whereas it is our decision to keep the complite definition for the abstract and the subsections' titles.
Finally, our idea is to publish our implemented simulator as soon as we will develop other radio access technologies (like mm-wave) and other functionalities for the SDN controller (e.g. multi-domain routing).